# Increasing Extractable Work in Small Qubit Landscapes

**DOI:** 10.3390/e25060947

**Published:** 2023-06-16

**Authors:** Unnati Akhouri, Sarah Shandera, Gaukhar Yesmurzayeva

**Affiliations:** 1Institute for Gravitation and the Cosmos, The Pennsylvania State University, University Park, PA 16802, USA; ses47@psu.edu (S.S.); yes.li.ju.9@gmail.com (G.Y.); 2Department of Physics, The Pennsylvania State University, University Park, PA 16802, USA

**Keywords:** open quantum systems, non-equilibrium dynamics, quantum thermodynamics

## Abstract

An interesting class of physical systems, including those associated with life, demonstrates the ability to hold thermalization at bay and perpetuate states of high free-energy compared to a local environment. In this work we study quantum systems with no external sources or sinks for energy, heat, work, or entropy that allow for high free-energy subsystems to form and persist. We initialize systems of qubits in mixed, uncorrelated states and evolve them subject to a conservation law. We find that four qubits make up the minimal system for which these restricted dynamics and initial conditions allow an increase in extractable work for a subsystem. On landscapes of eight co-evolving qubits, interacting in randomly selected subsystems at each step, we demonstrate that restricted connectivity and an inhomogeneous distribution of initial temperatures both lead to landscapes with longer intervals of increasing extractable work for individual qubits. We demonstrate the role of correlations that develop on the landscape in enabling a positive change in extractable work.

## 1. Introduction

The universe, even after nearly 14 billion years, is an out-of-equilibrium system. Within a large volume that was once hot and nearly homogeneous, a remarkable small-scale diversity of structures has developed. These sub-structures can be individually described as open systems, co-evolving under local interactions that generate non-equilibrium states with respect to an ambient average temperature. The reference temperature is defined on some larger scale and is generally also evolving. One way to characterize this type of system is through the evolution of non-equilibrium free-energy [1,2] of subsystems, which quantifies the amount of work that can be extracted if the subsystem is brought in contact with a bath at some temperature *T*. Subsystems for which extractable work increases with time are effectively extracting resources from the environment, to be used at a later time [3].

In this paper, we study the minimal ingredients required for a thermodynamic evolution that can result in the sustained generation of extractable work for subsystems. We address this question in two steps. First, we find the smallest quantum systems, made entirely of thermal qubits, that allow an increase in the extractable work for a subsystem. The small quantum machines prescribe a single-step evolution scheme for both the focal-subsystem qubit and a reference thermal qubit, allowing the former to exploit a change in the latter to achieve an increase in extractable work. Next, we consider co-evolving subsystems on small, closed landscapes of qubits. On the landscape, unitary evolution occurs in dynamically defined, rather than fixed, subsystem neighborhoods. The evolution conserves total energy while allowing some correlations to develop. The correlations then act as a resource that can be redistributed across the landscape. We find that both the quantum correlations that develop and the gradients in a *classical* notion of the reference temperature on the landscape affect how subsystems achieve an increase in extractable work. We study how properties of the qubit network, including the connectivity and initial state, affect the magnitude and persistence of steps for which the extractable work of a subsystem increases.

Within the framework of the thermodynamics of quantum systems, our work intersects with several developing directions in the literature, including studies on the role of extractable work for out-of-equilibrium systems, systems that share properties of life, [3,4,5,6], and on systems that utilize correlations and coherence for extractable work [7,8,9,10,11,12]. Because we consider finite systems, we can characterize the role of dynamics that are non-Markovian and not CP-divisble [13,14] in generating an increase in extractable work. The landscape approach can provide a new perspective on systems that resist rapid thermalization, as characterized by the evolution of the ensemble of subsystems. The use of a time-dependent Hamiltonian is the key distinction between the framework we consider and most work on understanding which closed systems have thermal subsystems. Both the eigenstate thermalization hypothesis [15,16,17] and the non-thermal, many-body localized phase of finite-sized quantum systems [18,19,20,21] rely on fixed, although disordered, Hamiltonians. In contrast, in the closed systems (the landscapes) that we study here, the Hamiltonian is time-dependent and does not have fixed eigenstates. Our work shares features with systems that show Hilbert-space fragmentation [22,23,24], by virtue of the total Hilbert space of our closed system being divided into dynamically independent subspaces. As is shown in recent literature these systems demonstrate non-thermalizing behaviour, which is crucial for our systems. As in the study of many-body localization [25], a careful characterization of behavior of these landscapes in the thermodynamic limit is an essential point for future work.

The landscape evolution can also be represented as quantum circuits and shares some features with random circuits studied in the literature [26,27,28]. However, in our work, we allow more complex connectivity and consider a conservation law that restricts the unitaries that may be applied and ensures that the landscape is thermodynamically closed. The dynamics we impose resembles that of the “thermodynamic compatibility" criterion of [29,30,31], but unlike that work, we do not impose a time-independent Hamiltonian or rely on a purely bipartite construction. Those restrictions lead to time-translation invariant, Markovian evolution, which is quite different for what occurs on the landscapes here.

Another set of related but distinct work is that on Quantum Cellular Automata (QCA). Those investigations (e.g., [32,33,34]) often focus on measures of complexity that can also be argued to be related to thermodynamically complex evolution, but in QCA, the update or evolution rule is deterministic and ultra-local, acting on a single qubit. In contrast, we consider random unitaries, restricted by the conservation law, acting on randomly selected multi-qubit subsystems. In addition, we want to study conditions that capture how finite resources on the landscape may be utilized and examine the statistics of incidents of increasing extractable work rather than some other measure of complexity.

In the remainder of the introduction, we define the framework we will use more formally. In Section 2, we define the smallest qubit machine that can lead to an increase in extractable work. In Section 3, we study the evolution of larger qubit landscapes. Contrasting the landscape results with the small machines illustrates the utility of correlations in obtaining positive extractable work, generalizing previous results [9,10].

### Defining the Rules of the Game: Initial States and Evolution

We are interested in constructing a system with no external thermodynamic sources or sinks, and one where the advantage, if any, of quantum dynamics may be tracked. To that end, we consider systems of *N* initially uncorrelated qubits beginning in mixed states described by thermal density matrices for each qubit,
(1)$$\rho_{\left(i\right)}=\left(1-p_{i}\right)\left|0\rangle \langle 0\right|+p_{i}|1\rangle \langle 1|\;,$$
where i=1,…,N. Here the orthonormal states |0〉 and |1〉 can be thought of as the ground state and excited state, respectively, with the free Hamiltonian for the *i*th qubit given by
(2)$${\hat{H}}_{0}^{\left(i\right)}=E_{0}^{\left(i\right)}\left|0\rangle \langle 0\right|+E_{1}^{\left(i\right)}|1\rangle \langle 1|\;.$$

This Hamiltonian serves to define several thermodynamic quantities. We set all $E_{0}^{\left(i\right)}=0$ for simplicity.

We consider $0<p_{i}<0.5$ so that the population fractions can be associated to a positive, finite temperature via the Gibbs distribution. Although we model small numbers of qubits with no explicit reference to external thermal baths, at least formally a temperature $T_{i}$ can be defined via
(3)$$k_{B}T_{i}=\frac{E_{1}^{\left(i\right)}}{\ln\left[\frac{1-p_{i}}{p_{i}}\right]}\;,$$
where $k_{B}$ is the Boltzmann constant.

The choice of initial state given in Equation (Equation 1) serves two purposes. First, it is a classical initialization that will allow us to track the role of quantum correlations that develop as the system evolves. Second, this choice anticipates the eventual goal of treating a distribution of many degrees of freedom in a large spatial volume, where local temperatures can be well-defined. The initial density matrix for the full system is the tensor product of the density matrices for each qubit at possibly different temperatures,
(4)$$\rho =\rho_{1}⊗\rho_{2}\ldots ⊗\rho_{n}\;.$$

For simplicity, we will work within the special case where the energy spacing of all qubits is identical, and then units are defined by $E_{1}^{\left(i\right)}=1$. In that case, the expectation value of the energy for each qubit is
(5)$$\langle E_{i}\rangle =\mathrm{Tr}\left[\rho_{i}{\hat{H}}_{0}^{\left(i\right)}\right]=p_{i}\;,$$
where ${\hat{H}}_{0}^{\left(i\right)}$ is the Hamiltonian associated with the *i*th qubit. The initial energy of the full system is the sum of the individual ensemble energies, $\langle E\rangle =\sum_{i}p_{i}$.

We will use the definition of the free Hamiltonian for the qubits to restrict the class of operations we will consider. Since we want to study the co-evolution of the qubits, as they trade thermodynamic quantities, we choose to work with unitaries that commute with the free Hamiltonian of the N-qubit system,
(6)$${\hat{H}}_{0}={\hat{H}}_{0}^{\left(1\right)}⊗1^{\left(N-1\right)}+1^{\left(1\right)}⊗{\hat{H}}_{0}^{\left(2\right)}⊗1^{\left(N-2\right)}+\cdots +1^{\left(N-1\right)}⊗{\hat{H}}_{0}^{\left(N\right)}$$
where $1^{\left(k\right)}$ represents an identity matrix acting on *k* qubits. This class of operations obey the energy conservation condition
(7)$$[\hat{U},{\hat{H}}_{0}]=0\;.$$

Often, interaction Hamiltonians are considered that do not commute with the Hamiltonian of the system and can result in unaccounted for sources of work and energy [35]. Restricting to unitaries that commute with the free Hamiltonian allows us to not only define a conservation law for the qubits but also allows us to isolate the thermodynamic effect of coherence and correlation that the focal qubit has with the other qubits on the landscape. Over the course of evolution the coherence can act as a resource that is transferred and shared between the qubits.

In the literature, a closely related choice of dynamics has been used to study fundamental limitations for quantum thermodynamics [36,37,38]. This class of thermal operations can be implemented by a dynamical map, Λ[.], acting on the focal system ($\rho_{f}$) as,
(8)$$\Lambda \left[\rho_{f}\right]={\mathrm{Tr}}_{B}\left[{\hat{U}}_{\mathrm{fB}}\left(\rho_{f}⊗\rho_{B}\right){\hat{U}}_{\mathrm{fB}}^{†}\right]\;.$$

Here, ${\hat{U}}_{\mathrm{fB}}$ is the energy-preserving unitary that commutes with the focal-system and bath Hamiltonian ${\hat{H}}_{\mathrm{fB}}={\hat{H}}_{f}⊗1_{B}+1_{f}⊗{\hat{H}}_{B}$, and $\rho_{B}$ is the bath or reservoir in a Gibbs state
(9)$$\rho_{B}=\frac{\exp^{-\beta H_{B}}}{Z_{B}}\;.$$

Owing to the restrictions on the evolution and the choice of factorized initial states, this map is Gibbs-preserving, does not generate coherence between energy levels, and is completely positive. Consequently the relative entropy, or the distance between qubits evolved with this map, monotonically decreases.

The evolution we consider is similar to the class of thermal operations in that the unitaries must commute with the free Hamiltonian of the full system, Equation (Equation 7). As a consequence, in the total density matrix of the qubits system, the two states with all the qubits either in the ground state, |00…0〉 or the excited state, |11…1〉, only evolve by separate U(1) rotations. The remaining states may be divided according to the energy subspaces E=1, …, E=N−1 with dimension ${}^{N}C_{1}$, ${}^{N}C_{2}$… ${}^{N}C_{N-1}$ respectively where ${}^{N}C_{M}=\frac{N!}{M!(N-M)!}$. Allowed unitary rotations can then be broken into a block diagonal form acting in subspaces of fixed energy,
(10)$$\hat{U}=\hat{U}\left(1\right)⨁{\hat{U}}_{E=1}⨁{\hat{U}}_{E=2}\ldots ⨁{\hat{U}}_{E=n-1}⨁\hat{U}\left(1\right).$$

This results in division of the full Hilbert space into dynamically independent subspaces. Under evolution the off-diagonal elements of any individual qubit evolve independently from the diagonal elements. Thus, single-qubit Gibbs states are mapped to new Gibbs states. From this unitary, Equation (Equation 10), we can derive the form of the dynamical map Φ[.] that drives the evolution of a focal qubit (*f*) evolving in an environment also made up of qubits (in the construction here, not a thermal bath) E,
(11)$$\rho_{f}\left(t+1\right)=\Phi_{t+1,t}\left[\rho_{f}\left(t\right)\right]={\mathrm{Tr}}_{E}\left[U.\rho_{fE}\left(t\right).U^{†}\right]$$
where *t* is the step index in the evolution. The explicit form of the superoperator acting on the vectorized form of the focal density matrix, $\overset{⇉}{\rho_{f}}=\left(\rho_{00},\rho_{01},\rho_{10},\rho_{11}\right)$, is
(12)$$\Phi_{t+1,t}=\left(\begin{array}{cccc} 1-\Delta_{} & 0\;\; & \;0\;\; & 1-{\tilde{\Delta }}_{}\\ 0 & \Gamma & 0 & 0\\ 0 & 0 & \Gamma & 0\\ \Delta_{} & 0 & 0 & {\tilde{\Delta }}_{} \end{array}\right)$$
where $\Delta_{}$, ${\tilde{\Delta }}_{}$ and Γ depend on the parameters of the unitary and the state of the environment qubits at time *t*. In structure, this map is identical to the one-qubit Davies map which drives the evolution for a qubit under thermal operations with a fixed Hamiltonian [39].

However, the maps considered in our work go beyond those generated by thermal operations, in that the range of the map parameters expands. This is due to the difference in the form of the environment we consider and in the type of states we apply the evolution on. Firstly, our environment, $\rho_{E}$, consists of qubits that were independently in contact with different reservoirs at different temperatures. The temperature inhomogeneity in the bath of the machine, as we will demonstrate below, facilitates generation of $\Delta W^{\mathrm{ex}}\geq 0$ under one application of the unitary extending the results in literature that relate the inhomogeneity in reservoir temperature with addition of new resource states [40].

Secondly, when we explore the development of $\Delta W^{\mathrm{ex}}\geq 0$ under co-evolution of multiple qubits on the closed landscape, the energy-preserving unitaries no longer act on uncorrelated states (except at the first step). As has been discussed in the literature [41,42], this can lead to individual qubit maps that are not completely positive (CP). Furthermore, as the qubits are co-evolving and the subsystems where unitary rotations act are chosen randomly, and the environment seen by any qubit changes at each step of the evolution. In summary, the inhomogeneity in environment temperature and the use of correlated states modifies the range of parameters in Equation (Equation 12) compared to those found in thermal operations.

While the map induces open-system evolution on individual qubits, the class of unitaries considered has a global conserved quantity. The total energy of the system is strictly conserved, so if $p_{i}$ indicates the initial population fraction of the *i*th qubit,
(13)$$\langle E\rangle =\sum_{i}p_{i}=\sum_{i}q_{i}$$
where $q_{i}$ is the population fraction of the excited state at any time t>0. In addition, we show below that the conservation law bounds the population fraction (and von Neumann entropy, and effective temperature) of any individual qubit to remain between the maximum and minimum of the qubits in initial state.

The energy of a system 〈E〉 can be combined with the von Neumann entropy *S* of a system described by the density matrix ρ, weighted by a reference temperature, *T*, to give the non-equilibrium free energy [2] of the system,
(14)F(ρ)=〈E〉−TS(ρ).

Since we require 〈E〉 for the whole system to be constant, there can be no source of free energy that is not entirely accounted for within the qubit system itself [35]. That is, the restriction to energy-conserving evolution helps to keep the systems considered here closed.

We will define the free energy for subsystems of various sizes, and take the reference temperature *T* to be determined by the state of some qubit(s) outside the subsystem but co-evolving within the closed system. Any excess in the free energy of a system described by density matrix ρ, above that of the same system in the reference thermal state $\rho_{\mathrm{th}}$ at *T*, defines the extractable work:(15)$$W^{\mathrm{ex}}=F\left(\rho \right)-F_{\mathrm{th}}\left(\rho_{\mathrm{th}}\right)\;.$$

We are interested in understanding dynamics that allow the extractable work to increase for a subsystem.

For a Gibbs-preserving process that takes $\rho_{\mathrm{th},0}$ to $\rho_{\mathrm{th},1}$, and the subsystem from $\rho_{0}$ to $\rho_{1}$, an increase in extractable work requires
(16)$$\Delta W^{\mathrm{ex}}=\left[F\left(\rho_{1},T_{1}\right)-F_{\mathrm{th}}\left(\rho_{\mathrm{th},1},T_{1}\right)\right]-\left[F\left(\rho_{0},T_{0}\right)-F_{\mathrm{th}}\left(\rho_{\mathrm{th},0},T_{0}\right)\right]>0\;,$$
where the reference temperature may evolve from $T_{0}$ to $T_{1}$. To understand the bounds on increasing the extractable work [3], it is helpful to keep in mind that the difference of free energies is proportional to a quantity that cannot decrease with time under positive evolution [43], the relative entropy, $D(\rho_{1}||\rho_{0})$, between the two states:(17)$$F_{1}-F_{0}=k_{B}T\ln2D(\rho_{1}\left|\right|\rho_{0})\;.$$

Kolchinsky et al. [3] recently discussed criteria on generic classical and quantum stochastic processes that lead to an increase in extractable work. Here, we will find the minimum requirements on a small, self-contained quantum system that includes a representative of the bath at temperature *T* such that $\Delta W^{\mathrm{ex}}>0$ can occur at least sometimes during the evolution of a subsystem. The conservation law we have imposed and the finite system size of the small machines both lead to restrictions in our in Section 2 results compared to the conclusions in [3]. On the other hand, the co-evolving systems on the landscape result in single-qubit dynamics that is more complex than the single-step, completely positive evolution of the quantum machines. While the evolution of the overall landscape is closed, few-qubit subsystems within the landscape most generally undergo non-unitary, non-Markovian evolution (relevant for work extraction via erasure [10]). In particular, we will be able to demonstrate the role of development of correlations between environment qubits in obtaining increases in extractable work. Our results extend recent work exploring the effect of environment-environment correlations on non-Markovianinty and memory effects in qubit systems [44,45,46].

We define the notion of interesting substructures by the following criteria:In small qubit machines, the occurrence of $\Delta W^{\mathrm{ex}}>0$ under application of a single dynamical map arising from an energy-preserving unitary on a focal qubit and reference thermal qubit.On the landscape, the length and distribution in time of intervals over which qubits exhibit $\Delta W^{\mathrm{ex}}>0$.

In the small qubit machines (next Section) we are able to affirmatively characterize athermal qubits (relative to a reference temperature *T*) as a resource which making the possible state space bigger and allows a positive change in extractable work. We numerically demonstrate that in a single-step evolution, a minimum of four qubits is needed to give a positive change in extractable work. Then, when qubits are replaced on a landscape (Section 3), an intermediate connectivity that is neither too restrictive nor too competitive is most conducive to generating steps with positive change in extractable work. On the landscape, we show how the correlations developed between the qubits lead to the development of additional sites with positive free energy gain.

## 2. Qubit Machines

In this section, we demonstrate that the smallest qubit machine that can lead to a positive change in extractable work for a focal qubit requires four qubits.

The *N*-qubit machine is defined by a unitary evolution in the class Equation (Equation 7) together with the initial state of N−1 qubits, which act as the environment for the focal qubit. Together, the unitary and the environment qubits define a stochastic map that transforms a *focal system* (or *actor*) qubit in state $\rho_{\mathrm{sys}}$ by coupling it to a *reference* state $\rho_{\mathrm{ref}}$ via a unitary of the type given in Equation (Equation 18). We schematically denote the choice of unitary by θ. It is easy to see that this is precise in the case of two qubits, where a generic unitary of the type we consider can be written
(18)$${\hat{U}}_{2Q}=\left(\begin{array}{cccc} 1 & 0 & 0 & 0\\ 0 & \cos\theta & e^{i\phi }\sin\theta & 0\\ 0 & -e^{-i\phi }\sin\theta & \cos\theta & 0\\ 0 & 0 & 0 & 1 \end{array}\right).$$

Here, it is sufficient to consider maps $\Phi (\rho_{\mathrm{sys}}|\rho_{\mathrm{ref}};\theta )$ defined in terms of the rotation angle, θ. Including the phases ϕ does not change the conclusions. On the larger Hilbert spaces of larger machines, there are additional rotation angles appearing in the allowed unitaries. We also label this more general set as θ. Figure 1 schematically illustrates the qubit machines of size N=2,3 and 4.

### 2.1. Two-Qubit Machines

First, consider two qubits initialized in thermal states as given in Equation (Equation 1), characterized by $p_{1}$ and $p_{2}$. The energy-conserving transformations contain a 2×2 block of rotations among the equal energy states (|01〉 and |10〉). This class of unitaries can be thought of as (partial) swapping operations, partially exchanging the population fractions in the |01〉 and |10〉 states. The total density matrix develops correlations reflected in the non-zero off diagonal elements under evolution. In Appendix A, we derive the conditions for some of those correlations to be quantum by computing the concurrence [47].

To determine whether a two-qubit system contains sufficient parameters and structure to describe an increase in extractable work, we evolve a generic actor qubit via the machine defined by the map $\Phi (.|\rho_{\mathrm{ref}};\theta )$. Applying the same machine to a qubit initialized at the reference temperature allows us to use the monotonicity of relative entropy under stochastic processes, D(ρ||σ)≥D(Φ(ρ)||Φ(σ)), to understand the requirements for achieving an increase in extractable work. Our results will then be a special case of those in [3], which considered more generic states and processes.

The transformations of the focal system, or actor, qubit and the reference qubit are
(19)$$\begin{array}{cc} \rho_{\mathrm{sys}} & \to \Phi \left(\rho_{\mathrm{sys}}|\rho_{\mathrm{ref}};\theta \right)=\rho_{\mathrm{sys}}^{\prime }\\ \rho_{\mathrm{ref}} & \to \Phi \left(\rho_{\mathrm{ref}}|\rho_{\mathrm{ref}};\theta \right)=\rho_{\mathrm{ref}}^{\prime }\;. \end{array}$$

This process is Gibbs preserving; both $\rho_{\mathrm{sys}}^{\prime }$ and $\rho_{\mathrm{ref}}^{\prime }$ are diagonal, and so the final reference temperature, $T^{\prime }$, is defined from $\rho_{\mathrm{ref}}^{\prime }$. Combining Equations (Equation 16) and (Equation 17), and the monotonicity of the relative entropy, shows that for a stochastic process that is Gibbs preserving, the reference temperature must go up ($T^{\prime }>T$) in order to have an increase in extractable work [3],
(20)$$\begin{array}{c} \Delta W^{\mathrm{ex}}=k_{B}T^{\prime }\ln2D(\rho_{\mathrm{sys}}^{\prime }\left|\right|\rho_{\mathrm{ref}}^{\prime })-k_{B}T\ln2D(\rho_{\mathrm{sys}}\left|\right|\rho_{\mathrm{ref}})\;. \end{array}$$

However, under a single qubit machine defined by $\rho_{\mathrm{ref}}$, the evolution of the reference temperature qubit is trivial, $\rho_{\mathrm{ref}}\to \rho_{\mathrm{ref}}^{\prime }=\rho_{\mathrm{ref}}$, and consequently the reference temperature remains unchanged. Unsurprisingly, a single qubit machine is not sufficient to characterize a scenario with an increase in extractable work.

### 2.2. Three-Qubit Machines

Now consider a set of three qubits, with identical energy levels for each qubit, and for simplicity $E_{0}=0$, $E_{1}=1$. Energy-preserving transformations now consist of two independent sets of rotations, one among the three states of total energy E=1 and the other in the three states of total energy E=2. For this case, we can consider the actor system to be either a subsystem with two qubits, or with one.

First, consider a two-qubit actor subsystem. In this case, the evolution for the subsystem is not Gibbs preserving and consequently a thermal state is not mapped onto a thermal $\rho_{1,\mathrm{th}}\to \rho_{1}^{\prime }\neq \rho_{1,\mathrm{th}}^{\prime }$. In many cases, as shown in [3], processes that are not Gibbs preserving can lead to an increase in extractable work, irrespective of the final temperature. This is because the system can be prepared in a thermal state according to the initial reference temperature and then
(21)$$\begin{array}{c} \Delta W^{\mathrm{ex}}=k_{B}T_{1}\ln2D(\rho_{1}\left|\right|\rho_{\mathrm{th},\;1})-k_{B}T_{0}\ln2D(\rho_{\mathrm{th},\;0}\left|\right|\rho_{\mathrm{th},\;0})=k_{B}T_{1}\ln2D(\rho_{1}\left|\right|\rho_{\mathrm{th},\;1}) \end{array}$$
which is positive definite as long as $D(\rho_{1}||\rho_{\mathrm{th},\;1})>0$. However, our three qubit example is a special, trivial case of this: if we use the third qubit to set the initial thermal state and reference temperature, then initializing the subsystem in the same thermal state, to track the evolution of the temperature under the same map, sets all three qubits to be at the same temperature. The evolution of the reference temperature is, thus, trivial and cannot result in an increase in extractable work.

Now, consider the case of a one-qubit actor subsystem as is shown in Figure 1b. Now, $\rho_{\mathrm{sys}}$ is our subsystem of interest and the initial reference temperature and thermal state is set by $\rho_{\mathrm{ref}}$. The third qubit given by the density matrix $\rho_{\mathrm{en}1}$ now acts as an enabler allowing non-trivial evolution for both the subsystem and the reference temperature qubit. The map that defines the evolution of the subsystem and the reference temperature is $\Phi (.|\rho_{\mathrm{ref}},\rho_{\mathrm{en}1};\theta )$. The evolved density matrices for the subsystem and the reference temperature under this map are given by
(22)$$\begin{array}{cc} \rho_{\mathrm{sys}} & \to \Phi \left(\rho_{\mathrm{sys}}|\rho_{\mathrm{ref}},\rho_{\mathrm{en}1};\theta \right)=\rho_{\mathrm{sys}}^{\prime }\\ \rho_{\mathrm{ref}} & \to \Phi \left(\rho_{\mathrm{ref}}|\rho_{\mathrm{ref}},\rho_{\mathrm{en}1};\theta \right)=\rho_{\mathrm{ref}}^{\prime }\;. \end{array}$$

As before, the one-qubit state after evolution, $\rho_{\mathrm{sys}}^{\prime }$ remains fully mixed, so the qubit undergoes a Gibbs-preserving process. Without loss of generality, either of the two remaining qubits (the environment qubits) may be used to set the initial reference thermal state and reference temperature. Here, we have chosen qubit 2 to set the reference temperature. For an evolution defined by a fixed sequence of unitary operators, the reference state and temperature after evolution is found by replacing the system qubit by a copy of the reference qubit; see Equation (Equation 22). If the reference qubit begins at a lower temperature than the remaining environment qubit ($T_{\mathrm{ref}}<T_{\mathrm{en}1}$), then generically its temperature can increase. However, this is not a sufficient condition for the extractable work to increase since $D(\rho_{\mathrm{sys}}||\rho_{\mathrm{ref}})$ decreases as $T_{\mathrm{ref}}$ increases. At the level of three qubits with the above defined unitary characterized by six rotation angles θ, and for the entire parameter space for the initial qubit population $0\leq p_{i}\leq 0.5$, we do not see a case for positive change in extractable work for randomized trials over the entire parameter space.

We have considered three identical qubits for simplicity, but one can check that allowing different level spacing while maintaining energy conservation does not generate a more flexible system. Consider three qubits with $E_{1}^{\left(1\right)}<E_{1}^{\left(2\right)}$ and $E_{1}^{\left(3\right)}=E_{1}^{\left(2\right)}-E_{1}^{\left(1\right)}$ (and all $E_{0}^{\left(i\right)}=0)$. If $E_{1}^{\left(2\right)}=2E_{1}^{\left(1\right)}$ there are three, two-dimensional subspaces of equal energy. For any other choice for $E_{1}^{\left(2\right)}$, there is a single two-dimensional energy subspace with energy $E_{1}^{\left(2\right)}$ plus six isolated states of different energies. This is the setup used by [48] to construct the smallest quantum refrigerator. If qubits one and two are initialized in thermal states at a cold temperature $T_{c}$, the state with qubit two in the ground state and qubit one in the excited state, |01〉, is more probable than the state |10〉. If the third qubit is initialized in a thermal state at a hotter temperature $T_{h}>T_{c}$, then |101〉 is more populated than |010〉, and an energy conserving operation that swaps the populations of those two states effectively cools qubit 1. Although this system cools one qubit at the expense of heating another, it does not generate processes that allow an increase in extractable work. If qubit 1 is considered the actor, or system, and the hot temperature qubit as the reference temperature, then the reference qubit is also cooled by the machine. Since the reference temperature goes down, the extractable work cannot increase. Consider instead taking qubit two as the actor (since it heats up) and $T_{h}$ as the initial reference temperature. But, the evolution of the reference temperature is found by initializing the second qubit at $T_{h}$ and performing the same rotation with the same environment qubits. In that case, the initial state begins with |010〉 more populated than |101〉 and so the machine that heats the system qubit cools the reference qubit. Again, the reference temperature goes down and no gain in extractable work can be achieved. If qubit three is the actor, the reference temperature is $T_{c}$, which cannot evolve under a machine running at $T_{c}$. In other words, small quantum refrigerators do not automatically generate increases in extractable work, and allowing different energy spacing does not change the conclusion that three qubits is insufficient to demonstrate $\Delta W^{\mathrm{ex}}>0$.

### 2.3. Four-Qubit Machines

Now consider four qubits, each with $E_{0}=0$, $E_{1}=1$. There are four-state sets with energy E=1 and E=3, and a six-state set with energy E=2. The same class of two qubit conditional swaps that was available in the three-qubit and two-qubit systems is allowed again for four qubits. Additionally, a new class of evolution is now allowed within the E=2 states, corresponding to simultaneous swaps of two-qubit pairs. For example,
(23)$$\begin{array}{cc} {\hat{U}}_{2\mathrm{pairs}}= & \left|0000\rangle \langle 0000|+\cdots +\cos\theta |1001\rangle \langle 1001\right|\\ \; & +\cos\theta |0110\rangle \langle 0110|-\sin\theta |1001\rangle \langle 0110|+\sin\theta |0110\rangle \langle 1001|\;, \end{array}$$
where the dots denote diagonal terms in the other basis states, is a simultaneous partial swap of two pairs of qubits. This, along with the presence of a fourth distinct qubit, expands the state space available for an individual qubit to explore under evolution. For a general rotation in the a four-qubit system, the set of fourteen rotation angles and three (two) distinct initial states of the machine qubits govern the accessible states of single (two) actor-qubit subsystem. The correlations can once again be read from the generation of off-diagonal terms that correspond to rotations between equal energy states.

For four qubits, the actor subsystem could consist of three, two or one qubit(s). For the case with a three qubit subsystem, while the evolution of the actor subsystem is non-trivial, the evolution of the reference temperature is once again trivial. Since the temperature is unchanging, once again there is no evolution of the system, and no change in the extractable work.

For the case of a two qubit actor subsystem, we can arbitrarily choose two out of the four qubits to define the initial state of the subsystem as $\rho_{\mathrm{sys}1}⊗\rho_{\mathrm{sys}2}$. The remaining two qubits and the unitary define the stochastic process and the reference temperature. We choose the third qubit to define the reference temperature. Then, the evolution of the subsystem and the reference temperature is $\Phi (.|\rho_{\mathrm{ref}},\rho_{\mathrm{en}1},\theta )$. The evolved density matrices for the subsystem and the reference temperature under this map are given by
(24)$$\begin{array}{cc} \rho_{\mathrm{sys}1}⊗\rho_{\mathrm{sys}2} & \to \Phi \left(\rho_{\mathrm{sys}1}⊗\rho_{\mathrm{sys}2}|\rho_{\mathrm{ref}},\rho_{\mathrm{en}1};\theta \right)=\rho_{\mathrm{sys}}^{\prime }\\ \rho_{\mathrm{ref}}⊗\rho_{\mathrm{ref}} & \to \Phi \left(\rho_{\mathrm{ref}}⊗\rho_{\mathrm{ref}}|\rho_{\mathrm{ref}},\rho_{\mathrm{en}1};\theta \right)=\rho_{ref}^{\prime }\;. \end{array}$$

Both the actor subsystem and the reference temperature undergo non-trivial transformations. Before discussing the scope of this system and process in extracting work, we will present the third alternative of a one qubit subsystem as is shown in Figure 1c. Now, $\rho_{\mathrm{sys}}$ is our subsystem of interest and the initial reference temperature and thermal state is set by $\rho_{\mathrm{ref}}$. The third and fourth qubits ($\rho_{\mathrm{en}1}$ and $\rho_{\mathrm{en}2}$) now act as enablers, allowing non-trivial evolution for both the subsystem and the reference temperature qubit. The map is now given by
(25)$$\begin{array}{c} \rho_{\mathrm{sys}}\to \Phi \left(\rho_{\mathrm{sys}}|\rho_{\mathrm{ref}},\rho_{\mathrm{en}1},\rho_{\mathrm{en}2};\theta \right)=\rho_{\mathrm{sys}}^{\prime }\\ \rho_{\mathrm{ref}}\to \Phi \left(\rho_{\mathrm{ref}}|\rho_{\mathrm{ref}},\rho_{\mathrm{en}1},\rho_{\mathrm{en}2};\theta \right)=\rho_{\mathrm{ref}}^{\prime }\;. \end{array}$$

For numerical simulation of such four-qubit systems, with both two-qubit subsystems as well as one qubit subsystems, we see positive change in extractable work under the allowed unitary evolution of the total system. Cases for $\Delta W^{\mathrm{ex}}\geq 0$ can be seen for four-qubit systems with four distinct initial temperature as well as three distinct initial temperatures (i.e., two qubits have the same initial state). To develop $\Delta W^{\mathrm{ex}}\geq 0$, under contractive CP maps, it is essential that the relative rise in temperature of the reference qubit must over come the fall in the relative entropy between the system and the reference qubit. At the level of three qubit systems, we could recover non-trivial evolution for the temperature of the reference qubit, but the trade-off between evolution of temperature and relative entropy was not sufficient. At the level of four-qubit systems we not only recover non-trivial evolution of reference qubit, but crucially, we also see that the rise in temperature overcomes the fall in relative entropy,
(26)$$\frac{T_{f}\left(\rho_{\mathrm{ref}}^{\prime }\right)}{T_{i}\left(\rho_{\mathrm{ref}}\right)}\geq \frac{D(\rho_{\mathrm{sys}}||\rho_{\mathrm{ref}})}{D(\rho_{{\mathrm{sys}}^{\prime }}||\rho_{{\mathrm{ref}}^{\prime }})}$$

This results in the development of $\Delta W^{\mathrm{ex}}\geq 0$ in four qubits. While we are yet to fully characterize the utility of the four qubit machines, our numerical investigation allows us to speculate that generation of $\Delta W^{\mathrm{ex}}\geq 0$ is related to the increase in the dimension of the equal energy subspace available for rotation.

## 3. Qubit Landscapes

In the previous sections, we found a minimal system for which $\Delta W^{\mathrm{ex}}\geq 0$ can occur in a subsystem. With this building block, we now explore the occurrence of $\Delta W^{\mathrm{ex}}\geq 0$ in a multi-step stochastic, closed evolution of a landscape of qubits. We perform an initial exploration of how the connectivity of the landscape (which defines the set of qubits on the landscape any one qubit can directly interact with), temperature variations in the initial state, and the choice of unitary evolution affects the evolution of temperature and $\Delta W^{\mathrm{ex}}$ on the landscape. The closed nature of the landscape makes it an example of co-evolving quantum systems [49,50].

We consider here landscapes of eight qubits labeled as $Q_{i}$ for i=1,2,…,8. The allowed interaction terms are defined by assigning a connectivity to the landscape. Unitary evolution can only occur in subsystems of connected qubits. Figure 2 shows the symmetric connectivities we consider. We also consider one asymmetric case, shown in Figure 3a, which we call the messenger-qubit system. On this landscape, we begin by choosing two subsystems of four qubits each–subsystem A and subsystem B. From each of these subsystems, a qubit is chosen to act as a messenger qubit. Between steps, these two qubits are exchanged between subsystems A and B. The messenger qubits ($Q_{4}$ and $Q_{5}$ in the figure) can directly interact with six qubits of the landscape, whereas the remaining six qubits can only directly interact with four other qubits of the landscape.

We initialize the system with thermal qubits, most at a uniform *cold* temperature and the rest at a *hot* temperature. We select a unitary from the family of four-qubit, energy conserving unitaries. For every step of evolution, two mutually exclusive four-qubit subsystems allowed by the connectivity are selected randomly. The subsystems each evolve under the unitary. This evolution can be visualized as a random quantum circuit, as shown in Figure 3b and Figure 4.

We compute the temperature associated with each qubit from its updated reduced density matrix, which after the evolution is still diagonal. Figure 5 shows the temperature associated with each qubit across 500 time steps, for each connectivity. In the case shown, each landscape was initialized with a single hot qubit and the chosen unitary is a simultaneous pair of qubit swaps in each subsystem, Equation (Equation 23). Even though this is a small, closed system, the top panel shows that it is large enough to show the expected trend to homogenization of temperature when the qubits are fully connected.

In contrast to Section 2, we no longer use the condition that the reference temperature and the focal qubit undergo the same evolution as this is not enforceable beyond one step. To frame the evolution of the out-of-equilibrium landscape in thermodynamic language requires introducing some notion of reference temperature, but this is not straightforward to define. There is an unambiguous notion of temperature for individual qubits, which always remain in Gibbs states, but not for the set of correlated qubits that make up the local environment [51,52,53,54]. Here we choose to define an effective reference temperature that encodes the information that would be classically accessible at a fixed time step by taking the average over the single-qubit temperatures in the multi-qubit environment. For example, for qubit 1, $Q_{1}$, on a landscape of *N* qubits
(27)$$T_{ref|Q_{1}}=\frac{1}{N-1}\sum_{i=2}^{N}\frac{1}{\log[\frac{1-p_{i}}{p_{i}}]}\;.$$

This choice of reference temperature allows us to isolate the role of quantum correlations in the thermodynamic evolution of the landscape. We comment further on this choice in Section 3.4.

Using reference temperatures computed from Equation (Equation 27) we compute an *effective* non-equilibrium free energy and then the *effective* change in extractable work, $\Delta W^{\mathrm{ex}}$, for each qubit compared to the qubit state and the reference temperature of its environment at the previous step:(28)$$\Delta W_{Q_{1}}^{\mathrm{ex}}=T_{ref|Q_{1}}^{\prime }D(\rho_{1}^{\prime }\left|\right|\rho_{\mathrm{th}}^{\prime })-T_{ref|Q_{1}}D(\rho_{1}\left|\right|\rho_{\mathrm{th}})$$
where $T_{ref|Q_{1}}^{\prime }$ and $T_{ref|Q_{1}}$ are the final and initial reference temperature for qubit $Q_{1}$ on the landscape, respectively. The state of the qubit $Q_{1}$ and of the thermal reference qubit at the reference temperatures is given by $\rho_{1}$ and $\rho_{\mathrm{th}}$, respectively. Under this definition for the *effective* free energy, we show that under evolution some qubits on the landscape develop a positive change in extractable work. This landscape does not have the machinery for maximal work extraction embedded in it, but the ensemble of the $\Delta W_{Q_{i}}^{\mathrm{ex}}$ contains the trade-offs associated with the restricted landscape evolution. This notion of extractable work is well suited to comparing out-of-equilibrium landscapes and delineating classical (effective temperature) effects from quantum correlation effects but should be interpreted with caution beyond this landscape setting. Figure 6 shows the change in extractable work for each qubit under the same evolution shown in Figure 5.

There are different ways of forming the subsystems that will interact together at any one step of evolution, restricted by the connectivity. Thus, qubits in any subsystem can utilize other qubits on the landscape as resources to explore a larger state space under the course of evolution. This is different from the evolution considered in Section 2, where there were no external resources a system could utilize, and in fact, it relaxes the need to choose pockets of four-qubit subsystems to find an increase in extractable work. Nevertheless, we will continue to use four-qubit subsystems since the class of energy-preserving four-qubit unitaries since they include the simultaneous swap of two qubit pairs used for the small machines (Section 2.3) as well as two-qubit and three-qubit rotations in the form of unconditional and conditional (partial) swaps.

In the rest of the section, we study how the development of positive change in extractable work for the qubits is affected by the following:The level of connectivity. This changes the number of ways any one qubit can directly interact with another qubit and generate correlations on the landscape. The network graphs for connectivities we consider are shown in Figure 2 and Figure 3.The degree of initial temperature variation on landscape. We consider initializing the landscape with between one and seven hot qubits.The type of unitary. We consider two qubit conditional partial swaps, two qubit unconditional partial swaps and simultaneous partial swaps of two different qubit pairs.

All of the closed landscapes result in diffusion of energy from the initially hot qubit in the system, so that the qubits evolve through a range of temperatures and do not reach a mean temperature. In Section 3.4 we contrast these systems with two types of open landscapes: a fully Markovian collisional model and evolution under our landscape rules but with correlations thrown away at each step. The comparison illustrates how closed evolution contains temperature variations and correlations that generate instances of $\Delta W^{\mathrm{ex}}\geq 0$.

### 3.1. Results: Connectivity and Initial Temperature Inhomogeneity

To study the efficiency of diffusion of energy under different levels of connectivity, we ran 100 evolution consisting of 500 steps each, for the 3 levels of connectivity shown in Figure 2, starting with 1 hot qubit and 7 cold qubits. For each of the 100 evolutions, we chose a different unitary within the class of simultaneous two-qubit swaps and then used it throughout the evolution. The left panel of Figure 7 averages over all evolution (and all qubits at the same temperature) for each connectivity to show the percentage of steps for which a qubit that starts as cold (hot) develops $\Delta W^{\mathrm{ex}}\geq 0$ in a run of 500 steps. The mean number of cases for positive change in extractable work for a cold (hot) qubit on the landscape is smallest (largest) for full connectivity. The large number of times the hot qubit experiences $\Delta W^{\mathrm{ex}}>0$, corresponds to greater energy diffusion and faster thermalization (although energy dissipation and extractable work are not necessarily maximized for the same conditions [55]). However, for the qubits that begin in the cold state, the hot qubit is ultimately the resource that allows them to have $\Delta W^{\mathrm{ex}}\geq 0$ at some times. Although the difference between connectivities is not large, Figure 7 suggests that a landscape with restricted connectivity fares better in terms of generating the conditions for the initially cold qubits of the landscape to exhibit $\Delta W^{\mathrm{ex}}\geq 0$ persistently under evolution.

Some general features are seen across the different types of connectivity, including diffusion of energy and a decrease in the magnitude of $\Delta W^{\mathrm{ex}}$ as the system evolves. For the qubit initialized in the hot state, the reference temperature provided by the landscape necessarily goes up under the first stages of evolution. This is not the case for the cold qubits. Thus, the number of steps where a positive change in free energy develops for the hot qubit is more than that for cold qubits for any level of connectivity. The cold qubits are initially at the maximum distance from the hot qubit. Over the course of evolution, the magnitude of free energy decreases since any closed evolution overall decreases the distance between the states.

To study how the level of inhomogeneity of the initial *cold* versus *hot* temperature distribution on the landscape affects the thermodynamics, we evolved landscapes with different numbers of initially hot qubits. As the fraction of qubits that start out hot is increased, the total positive change in extractable work from the landscape decreases. The right panel of Figure 7 shows the total positive extractable work, normalized by the initial mean temperature of the landscape, summed over the entire landscape for the three different symmetric connectivities, under 500 steps. Although the connectivity is a comparatively small effect, the total extractable work for a given fraction of hot qubits is largest for the landscape with full connectivity. The decrease in the total extractable work, as a function of the number of hot qubits, is exponential for all connectivities.

From the results so far, it is clear that neither maximizing the of steps with $\Delta W^{\mathrm{ex}}\geq 0$ or the total magnitude $\frac{\Delta W^{\mathrm{ex}}}{\langle T\rangle }>0$ is the right way to pick out landscapes that are slow to thermalize. After confirming that varying the number of qubits directly interacting gives results consistent with the effects of connectivity, we will instead look at persistence measures (in Section 3.3).

### 3.2. Results: Varying the Unitaries

We now consider the effect of three different classes of unitaries which correspond to three classes of possible correlations that can be generated in the system:Simultaneous swap of two qubit pairs generated by interaction of the class ${\hat{H}}_{4Q}⊗1_{4}+1_{4}⊗{\hat{H}}_{4Q}$, where ${\hat{H}}_{4Q}$ is a four-qubit interaction in the energy subspace E=2, with two pairs of qubit simultaneously being swapped. A representative unitary of this type is given in Equation (Equation 23).Conditional two-qubit swap given by ${\hat{H}}_{4Q}⊗1_{4}+1_{4}⊗{\hat{H}}_{4Q}$, where ${\hat{H}}_{4Q}$ is a four-qubit interaction that swaps populations between two qubits of a four-qubit system. For example, the rotation generated by the Hamiltonian interaction |1000〉〈0100|+h.c.Unconditional two-qubit swap given by ${\hat{H}}_{2Q}⊗1_{6}+1_{2}⊗{\hat{H}}_{2Q}⊗1_{4}+1_{4}⊗{\hat{H}}_{2Q}⊗1_{2}+1_{6}⊗{\hat{H}}_{2Q}$ where ${\hat{H}}_{2Q}$ is a two qubit interaction that swaps populations. For example the rotation in ${\hat{H}}_{2Q}$ is generated by a Hamiltonian interaction of the type |10〉〈01|+h.c.

We ran 50 trials consisting of 500 steps each for a landscape with full connectivity and one initial hot qubit under the different types of unitary. Within each class, a rotation angle θ was chosen at random at the initialization stage. Figure 8 shows the distribution for the percentage of steps for which a qubit that starts as cold (hot) develops $\Delta W^{\mathrm{ex}}\geq 0$ out of 500 steps, over 50 trials. The unitaries that involve fewer qubits or restricted rotations in energy subspaces slow the rate of diffusion from the hot qubit and subsequently lower the number of times the hot qubit experiences $\Delta W^{\mathrm{ex}}>0$ in the 500 evolution steps. This is similar to the effect of restricted connectivity between qubits in Figure 7a. The slow rate of diffusion of the hot qubit also results in a larger number of times the cold qubits experience $\Delta W^{\mathrm{ex}}>0$ in the 500 evolution steps. This is because the landscape resists approaching the state where the mutual information between the qubits is the highest or where locally all qubits look similar. The state where all qubits become identical has no utility for extractable work on a closed landscape because the relative entropy between the states and thermal state of environment approaches zero. The other extreme, where qubits do not interact and evolve, is also of no utility because the change in extractable work would be zero. Therefore, some but not all correlations in the environment are a key resource for the generation and persistence of $\Delta W^{\mathrm{ex}}>0$, as we discuss below.

### 3.3. Results: Persistence of $\Delta W^{\mathrm{ex}}\geq 0$

In order to characterize the persistence of $\Delta W^{\mathrm{ex}}\geq 0$ on the landscape, we next examine the length of the intervals over which a qubit continuously achieves a positive change in extractable work.

To investigate this, we ran 100 trials of 500 steps for four different connectivities shown in Figure 2 and Figure 3. In all the trials, the landscape was initialized with one hot qubit and seven cold qubits. For the messenger subsystem, the hot qubit and one cold qubit were chosen as the messenger qubits since they exhibited more instances of $\Delta W^{\mathrm{ex}}\geq 0$ as compared to a case where both the messengers initially chosen are cold. Note that this step of choosing which qubits for the two subsystems act as the messenger completely eliminates any randomness in the sequence in which the qubits interact in the messenger-qubit system as is seen in Figure 3b. This is not the case for connectivity seven and six where there is randomness in the way subsystems that interact together is chosen (see Figure 4). Keeping all other parameters fixed—initial conditions and form of unitary for evolution—allowed us to highlight the effect of connectivity of landscape and subsequently the randomness in choosing the subsystems that interact together.

Figure 9a shows a log-linear scale histogram for the instances of positive change in extractable work grouped by the intervals, for all qubits on the landscape for the three connectivities. On the fully connected (connectivity seven) landscape, the majority of instances when $\Delta W^{\mathrm{ex}}\geq 0$ are short-lived, with interval length ≤ 40 steps. All landscapes with restricted connectivity are order-of-magnitude more long-lived, with interval length ≥ 20 steps and instances. The connectivity-six landscape and the messenger-system shows comparable instances of long-lived (≥40 steps) intervals of $\Delta W^{\mathrm{ex}}\geq 0$ compared to the more restricted five and less restricted seven connectivity landscape. In fact, the longest-lived interval was found in the connectivity-six system of ≥40 steps. Between the range of interval length 10–40 consecutive steps, connectivity five dominates in the number of instances.

To study the evolution of these long-lived $\Delta W^{\mathrm{ex}}\geq 0$ on the landscape, we also looked at when in the evolution the long-lived intervals occur. In Figure 9b, all three landscapes have instances of $\Delta W^{\mathrm{ex}}\geq 0$ even at early times due to the initial difference in temperature among qubits. However, it is now clear that the fully connected landscape has more instances of $\Delta W^{\mathrm{ex}}\geq 0$ earlier in the evolution (≤10%), but the sparsely connected landscapes eventually develop persistent and more long-lived $\Delta W^{\mathrm{ex}}\geq 0$. The landscape of connectivity six shows more cases of long-lived $\Delta W^{\mathrm{ex}}\geq 0$ than either the more (connectivity seven) or less (five) connected examples. The later-time behavior contrasts the homogenizing fully connected landscape with the more sparsely connected systems that continue to exchange athermality and evolution in dynamically disconnected sectors in the Hilbert space as a resource.

### 3.4. Contrast with Thermalizing Landscapes

We now contrast the landscapes studied in this work with related systems that display Markovian evolution and rapid thermalization [56]. First, we consider a model where each of the qubits is initialized as for the landscape but evolves via collision with bath qubits at a fixed temperature, set to the average initial temperature of the landscape. After each collision, any correlations developed with the bath are thrown away and the state of the bath qubit is reset for the next evolution step. The qubits undergo a Markovian evolution which has been studied as a microscopic model of thermalization [57].

The results are shown in Figure 10, compared to the full non-Markovian evolution of identically initialized qubits on landscapes of several connectivities. In the collisional case each qubit quickly reaches the average reference temperature with no instance of $\Delta W^{\mathrm{ex}}\geq 0$ ever developing. Our results share the key feature observed in previous studies [10]: for non-Markovian evolution only, the amplitude of extractable work shows an oscillatory behaviour, indicating a “revival” of work extraction. The plot illustrates the utility of non-Markovian evolution in developing positive changes in extractable work. This exercise further indicates how the analysis might change if a notion of temperature that folded in the correlations, as in ref. [53], was used.

To further illustrate the role of correlations [58], we consider evolution under our landscape rules where all correlations are thrown away at each step. That is, the unitaries always act on uncorrelated, diagonal density matrices for the total system. Figure 11 shows the comparison between dynamics with and without erasing correlations. The unitary for both the landscapes at all steps of the evolution is identical and the evolution occurs in the same set of randomly selected four-qubit subsystems. On the landscape with correlation, all the correlations developed in the full eight-qubit density matrix is retained. On the landscape without correlation, one evolution step corresponds to application of the density matrix followed by erasure of any off diagonal terms developed. From the remaining diagonal density matrix, we compute the updated state of each qubit and form the uncorrelated new density matrix which undergoes subsequent evolution. Any correlation formed between the qubits is lost and the landscape, at each step, is factorized. Note, in this setup, while the correlations are lost, the total energy is still conserved at each step. In this way, this landscape is energetically closed but is open in reference to entropy by continually losing mutual information.

We plot the change in ambient temperature (normalized by the average temperature on the landscape) for the cold qubit at that step, the change in relative entropy and the change in extractable work normalized by the average temperature on the landscape. The plot indicates that the amplitude of the change of the quantities under evolution without correlations shows a decreasing trend. When correlations are kept, fluctuations at late times can be as large as at early times which enables relatively large values of $\Delta W^{ex}>0$ later in the evolution. A closer look at the fluctuations indicates that on the landscape with correlations, the change in extractable work can be positive even when the change in temperature is negative. These features show the importance of correlations both among environment qubits and between environment and system qubits for generating positive change in extractable work. In comparison to the fully Markovian evolution where each qubit individually interacts with a fixed thermal qubit bath that is reset at each step in Figure 10, the landscape in Figure 11 Panel (b) benefits from some effect of correlations by allowing the unitary to be updated by the state of each qubit co-evolving on the landscape. Consequently, unlike the fully Markovian landscape, the state of the environment corresponding to each qubit is not reset but is updated simultaneously. Therefore, the plot shows some oscillation in the decay in $\Delta W^{\mathrm{ex}}\geq 0$, but this landscape still does not allow harnessing the full power of correlations available on the landscape shown in Panel (a). Our future work will characterize these correlations and their utility further more by studying the effect they have on the parameters ($\Delta ,\tilde{\Delta }$ and Γ) of the ensemble of dynamical maps (see Equation (Equation 12)) that evolve the landscape.

## 4. Discussion and Conclusions

We have constructed a small, closed, thermal system of four identical qubits that fully parameterizes an actor qubit together with a machine activated to allow an increase in extractable work for the actor when the reference temperature evolves via the same machine. Using four-qubit subsystems as a building block, we defined landscapes of eight co-evolving qubits with several parameters that can be controlled to explore the thermodynamic evolution of correlated subsystems. Varying the connectivity, the initial temperature distribution, and the number of qubits directly interacting at each step, we explored the frequency of evolution steps where the change in *effective* extractable work is positive, $\Delta W^{\mathrm{ex}}>0$, for any individual qubit and the total amount of $\Delta W^{\mathrm{ex}}/\langle T\rangle >0$ for all qubits on the landscape. Within the landscape, where the qubits that interact are randomly chosen at each time step, states develop that will eventually act as resources for neighboring qubits.

We saw that the number of steps where the change in extractable work was positive for the initially cold (hot) qubit was the smallest (largest) for the landscape with full connectivity. The total positive change in extractable work was largest on a landscape with full connectivity but upon increasing the fraction of initially hot qubits on the landscape the total, normalized, change in extractable work for each connectivity decreases exponentially. We found that unitaries that are more restricted in the energy subspace and number of qubits they connect result in a higher frequency of steps with a positive change in extractable work for the qubits initialized cold, which is consistent with the connectivity results. A higher frequency of $\Delta W_{\mathrm{ex}}>0$ events for a qubit initialized hot was correlated with the approach toward a more homogeneous temperature across the landscape rather than the development of any particularly interesting subsystems. On the other hand, a higher frequency of $\Delta W_{\mathrm{ex}}>0$ for cold qubits corresponded to a less thermalized landscape. We also looked at the endurance, through many steps, of $\Delta W^{\mathrm{ex}}>0$ for individual qubits. The distribution of number of steps in intervals for which a qubit exhibits $\Delta W^{\mathrm{ex}}>0$ shows that more sparsely connected landscapes produce more instances of and longer intervals over which $\Delta W^{\mathrm{ex}}>0$ at each step.

These results suggest that landscapes with somewhat restricted connectivity, and therefore restrictions on evolution within the disconnected sectors of the full Hilbert space, are better at developing a richer (less thermalizing) structure of sites with positive change in extractable work. It would interesting to understand how this result is connected with other studies showing the benefits of sparsely connected networks for other tasks, which has been observed in a variety of contexts in the literature (see, for example, [24,59,60] and references therein for the spectrum of such results).

However, the results for exploring the development of $\Delta W^{\mathrm{ex}}>0$ on non-equilibrium landscapes relied crucially on defining an effective reference temperature. The definition used here was motivated by an attempt to isolate the classical part effects by averaging the well-defined single qubit temperatures, Equations (Equation 27) and (Equation 28). Recent work [53,54] has suggested other possible prescriptions that fold in the effects of correlations between the qubits. It would be interesting to investigate how these prescriptions affect our conclusions or provide a different insight into the role of quantum correlations.

This exploratory study is restricted in utility by the small size of the landscape considered. In particular, the difference observed between different levels of connectivity was small. However, the results do qualitatively point the way toward measures of how well a closed thermodynamic landscape can do at producing thermodynamically rich subsystems. We have shown how sub-classes of unitary and connectivity can affect development of a locally defined $\Delta W^{\mathrm{ex}}>0$, but we have not explored the protocols for work extraction. Quantitatively defining physically consistent thermodynamic measures on larger landscapes, as well as considering work extraction protocols, would be helpful in more fully understanding out-of-equilibrium landscapes.

## Figures and Tables

**Figure 1 entropy-25-00947-f001:**
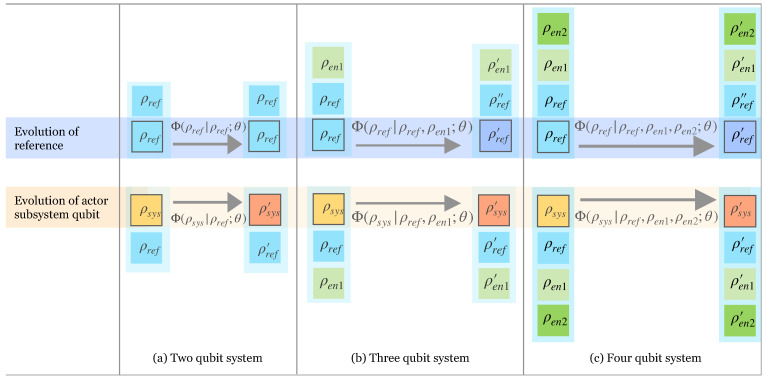
Pictorial representation of the action of qubit machines, where individual square blocks represent thermal qubits with density matrix $\rho_{i}$ as given in Equation (Equation 1). Each collection of boxes highlighted in blue contains both a block with a black border, representing the qubit that evolves under the stochastic process (the “actor” that activates the machine), as well as the qubits that contribute to the definition of the process (the machine). The map for evolution is denoted by $\Phi (.|\rho_{\mathrm{ref}},\rho_{\mathrm{en}1},\rho_{\mathrm{en}2},\ldots ;\theta )$ defined by the initial density matrices for the qubits in the machine and the unitary operation that couples the qubits, labeled by θ. The top half of the diagram shows the evolution of the reference temperature qubit $\left(\rho_{\mathrm{ref}}\right)$, and the bottom shows the evolution of the qubit system of interest, the “actor” qubit $\left(\rho_{\mathrm{sys}}\right)$. The other diverse qubits, represented by $\rho_{\mathrm{en}}$ in the machine, *enable* non-trivial transformations. Three qubits are required for a non-trivial evolution of the reference temperature. Four are required for a subsystem that has an increase in extractable work after evolution.

**Figure 2 entropy-25-00947-f002:**
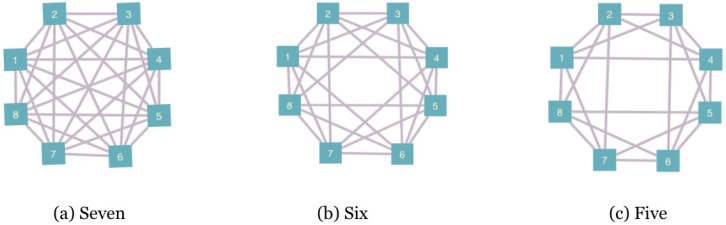
Symmetric connectivities considered on the landscape of eight qubits. The nodes represent the qubits and the links connect the qubits to the others they can directly interact with under unitary evolution.

**Figure 3 entropy-25-00947-f003:**
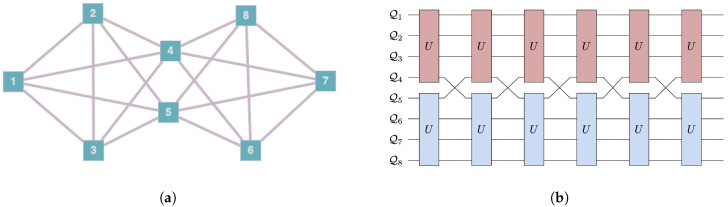
Connectivity (**a**) and circuit diagram (**b**) (for the first six steps of the evolution) for the messenger-qubit system. The two subsystems of size four that initially interact together are {$Q_{1},Q_{2},Q_{3},Q_{4}$} and {$Q_{6},Q_{7},Q_{8},Q_{5}$}, but $Q_{4}\;\mathrm{and}\;Q_{5}$ are then exchanged so that at the next step, the subsystems that interact together are {$Q_{1},Q_{2},Q_{3},Q_{5}$} and {$Q_{6},Q_{7},Q_{8},Q_{4}$}. Compared to the connectivities shown in Figure 2, the messenger-qubit system has asymmetric structure since the messenger qubits can participate in interactions with six other qubits of the landscape, while the members of subsystems only interact with four qubits on the landscape.

**Figure 4 entropy-25-00947-f004:**
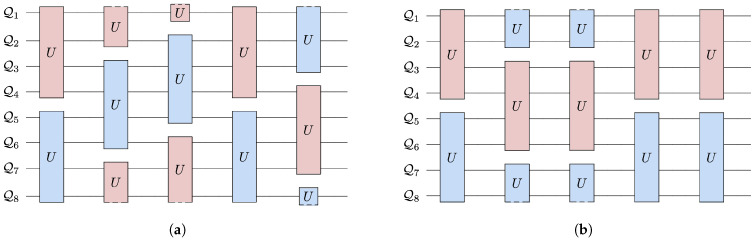
Quantum circuit for the first six steps of an example eight-qubit landscape with connectivity six (**a**) and five (**b**). The colors denote the four-qubit subsystem grouping that is randomly chosen to undergo a unitary evolution. We use a periodic boundary so that the first qubit, $Q_{1}$, is connected to last qubit, $Q_{8}$. The degree of connectivity manifests in the number of possible groupings. Panel (**a**) shows the four possible groupings allowed for connectivity six, and (**b**) shows that more restricted connectivity results in fewer possible groupings. In the paper we also consider the connectivity seven, full connectivity for an eight-qubit landscape, which allows grouping any four qubits into a subsystem.

**Figure 5 entropy-25-00947-f005:**
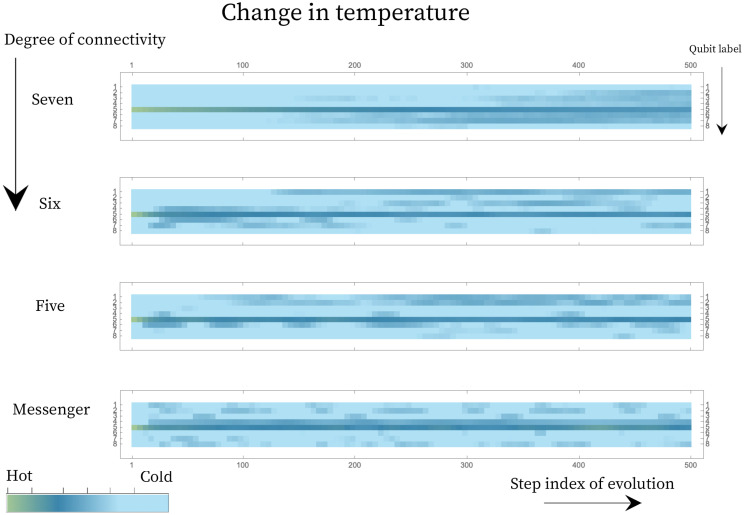
Evolution of the temperature of the 8 qubits under 500 steps in energy subspace E=2 for an angle θ=0.1, under different levels of connectivity. The hot qubit starts with a population fraction of $p_{h}=0.4$, and the colder qubits start at a population fraction of $p_{c}=0.2$. The diffusion for a connectivity of seven follows a more gradual trend towards looking homogeneous, whereas for the landscape of connectivity five and six, and the messenger system, pockets of hot and cold regions develop on the landscape.

**Figure 6 entropy-25-00947-f006:**
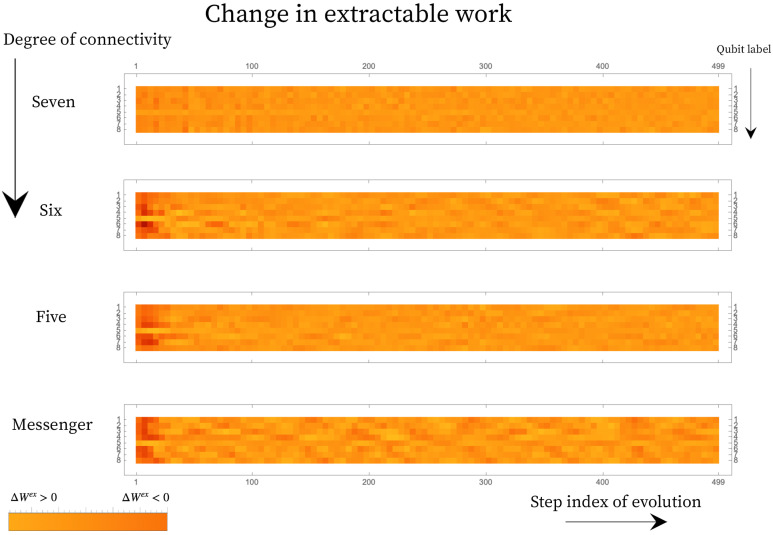
The change in extractable work from each qubit across 500 steps corresponding to the landscapes shown in Figure 5. The change in work is computed between two consecutive steps and thus a positive change corresponds to change with respect to the previous step. Fully connected landscape of connectivity seven shows dilution of $\Delta W^{\mathrm{ex}}\geq 0$ pockets, while the restricted connectivity show persistent instances of $\Delta W^{\mathrm{ex}}\geq 0$. Furthermore, the restriction in connectivity results in slow diffusion of energy from the hot qubit onto the landscape resulting in higher magnitude for extractable work early on on the landscape (as is depicted by the colors of the swatch).

**Figure 7 entropy-25-00947-f007:**
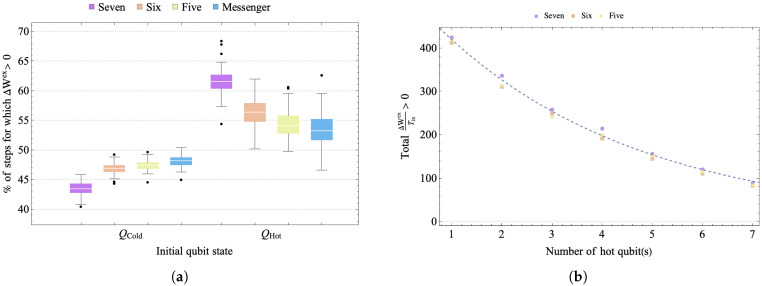
(**a**) Box plots showing the distribution in the percent of steps for which $\Delta W^{\mathrm{ex}}\geq 0$ for the qubits starting cold and hot, under 500 steps for the different levels of connectivity, for 100 trials. (**b**) Total $\frac{\Delta W^{\mathrm{ex}}}{\langle T\rangle }>0$ as a function of the fraction of hot qubits on the landscape, where 〈T〉 is the average initial temperature on the whole landscape. The fit for the plot for the degree of connectivity six is proportional to $e^{-0.25x}$, where *x* is the initial number of hot qubits on landscape.

**Figure 8 entropy-25-00947-f008:**
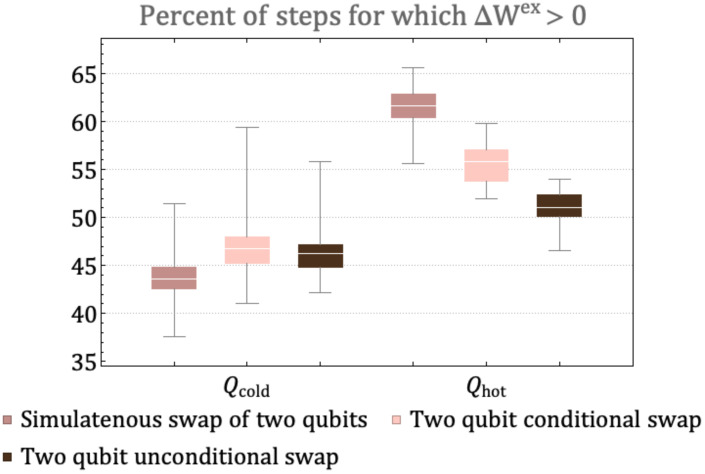
The distribution, over 50 trials, of the percent of steps for which $\Delta W^{\mathrm{ex}}\geq 0$ for a qubit starting of as cold and hot. All landscapes have connectivity seven but undergo evolution via different types of unitary rotations: simultaneous swap of two pairs of qubits, two-qubit conditional swap and two-qubit unconditional swap. A random unitary belonging to the class was chosen at the initialization step and then applied to each landscape in random subsystems for 500 steps.

**Figure 9 entropy-25-00947-f009:**
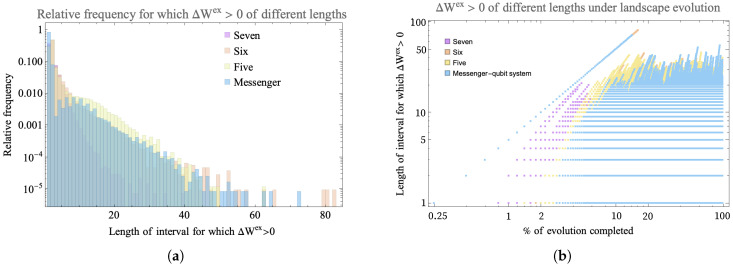
(**a**) Log-linear scale histogram (normalized to relative frequency) of different length intervals for which the qubit consecutively exhibits $\Delta W^{\mathrm{ex}}\geq 0$. The histogram shows data for 100 trials of 500 steps each for three types of connectivity: fully connected (seven), connectivity six, five and a messenger-qubit system where two subsystem of size four make up the landscape and under each iteration a messenger qubit is exchanged between the subsystems. The landscape is initialized with a hot qubit and seven cold qubits. The horizontal axis shows the number of steps in the interval over which $\Delta W^{\mathrm{ex}}\geq 0$ persists and the vertical axis shows the percent of total steps that occur within intervals of that length. (**b**) Log–log scale plot showing development of $\Delta W^{\mathrm{ex}}\geq 0$ of different lengths under 500 steps for 100 trials for three types of connectivity. Up until 10% of the evolution, the fully connected landscape with connectivity seven shows more instances of $\Delta W^{\mathrm{ex}}\geq 0$ than the other two landscapes, but in the later part of evolution, sparsely connected landscapes—connectivity six and messenger-qubit system—dominate by showing that more instances of connected landscape show more instances of $\Delta W^{\mathrm{ex}}\geq 0$ that are long-lived.

**Figure 10 entropy-25-00947-f010:**
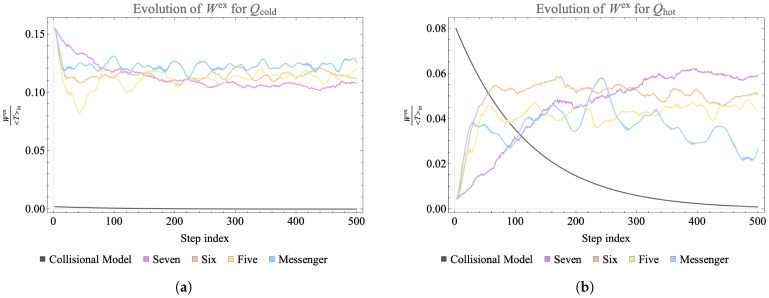
Time evolution of the extractable work divided by mean initial temperature of the landscape, $\frac{W^{\mathrm{ex}}}{{\langle T\rangle }_{in}}$, for a cold (**a**) and hot (**b**) one-qubit subsystem on the landscape. The dark gray line shows the extractable work from a qubit on a landscape that undergoes subsequent collisions with qubits at the initial mean temperature of the landscape. The purple, orange, yellow and blue lines show the time evolution of normalized extractable work of one-qubit subsystems on closed landscapes where the allowed interactions occur in subsystems chosen under several different connectivity constraints as indicated in the legend. In the collisional model there is a monotonic decrease in extractable work, while in all three closed landscapes revivals of work extraction can be observed.

**Figure 11 entropy-25-00947-f011:**
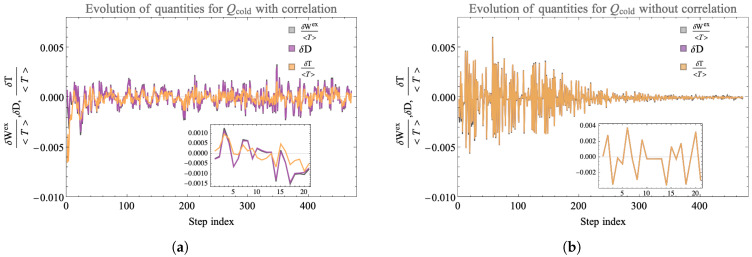
Here, we show the evolution of the change in the ambient temperature, Equation (Equation 27), the change in relative entropy and change in extractable work for an initially cold qubit that is evolving on a landscape with and without correlations kept. Panel (**a**) shows values for the landscape where all the correlations are retained. Panel (**b**) shows the evolution when any correlations formed are erased at each step. We see that on a landscape where correlations are erased, all three parameters for $\Delta W^{\mathrm{ex}}>0$ show an overall diminishing trend whereas for the closed landscape with correlations, the generation of $\Delta W^{\mathrm{ex}}>0$ persists.

## Data Availability

The Mathematica notebooks and datasets used and/or analysed during the current study is available from the corresponding author on reasonable request.

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
