# Peer review of "Increasing Extractable Work in Small Qubit Landscapes"

_entropy, 2023, doi:10.3390/e25060947_

Round 1

Author Response

We thank the reviewer for their constructive comments. 

Reviewer 2 Report

This work considers unitary dynamics of a small set of qubits prepared at different populations ("temperature"). The authors show that it's possible to create instances of "extractable work" with a four-qubit setup connected to another set of (four) qubits that act as the "environment".

I think that this is an interesting paper and deserves to be published. There are some issues that the authors should probably comment and/or elaborate on.

I have actually done some work on both connected quantum harmonic oscillators and spin qubits, where one of them is the "system" and the N-1 other form a "background". The QHO system can be solved exactly for a large number of oscillators which makes it possible to study the large-N limit. It is well-known (or at least expected) that these unitary models display re-entrant dynamical behavior such that energy initially inserted into the system will spread in the background, but will eventually come back to the system (partially), as these models do not thermalize. Thus, it may not be surprising that in the present case of 8 qubits there are oscillations that lead to an effectively higher "temperature" (population). Can the authors comment on this?

Quantum thermodynamics is still a developing field, and it is not clear to me that temperature is a well-defined quantity for an instantaneous population of a qubit. There are various suggestions how to define T for non-equilibrium (non-Gibbsian) states, see, e.g., https://arxiv.org/abs/2105.11915 for recent work. This is an important point because the "non-equilibrium free energy" cannot be well-defined if T is not. This raises a question whether or not one can even define "extractable work" in a consistent way. Is the effective free-energy difference defined by the authors physically consistent or meaningful, and how could actual physical work be extracted from it? I think the authors need to critically discuss these points.

Author Response

We thank the reviewer for taking the time to review our work and for providing constructive comments.  
